# Effects of Dietary Protein Intake on Cutaneous and Systemic Inflammation in Mice with Acute Experimental Psoriasis

**DOI:** 10.3390/nu13061897

**Published:** 2021-05-31

**Authors:** Tanja Knopp, Tabea Bieler, Rebecca Jung, Julia Ringen, Michael Molitor, Annika Jurda, Thomas Münzel, Ari Waisman, Philip Wenzel, Susanne Helena Karbach, Johannes Wild

**Affiliations:** 1Center for Thrombosis and Hemostasis (CTH), University Medical Center Mainz, 55131 Mainz, Germany; tanjaknopp@uni-mainz.de (T.K.); tbieler@students.uni-mainz.de (T.B.); rebeccajung@uni-mainz.de (R.J.); Julia.Ringen@unimedizin-mainz.de (J.R.); Michael.Molitor@unimedizin-mainz.de (M.M.); AnnikaKristin@gmx.de (A.J.); wenzelp@uni-mainz.de (P.W.); karbasu@uni-mainz.de (S.H.K.); 2Center for Cardiology—Cardiology I, University Medical Center Mainz, 55131 Mainz, Germany; tmuenzel@uni-mainz.de; 3German Center for Cardiovascular Research (DZHK)—Partner Site Rhine-Main, 55131 Mainz, Germany; 4Institute of Molecular Medicine, University Medical Center Mainz, 55131 Mainz, Germany; waisman@uni-mainz.de; 5Focus Program Translational Neurosciences, University Medical Center Mainz, 55131 Mainz, Germany; 6Research Center for Immunotherapy, University Medical Center Mainz, 55131 Mainz, Germany

**Keywords:** protein diet, psoriasis, inflammation, psoriasis-like skin disease, imiquimod

## Abstract

Background: Psoriasis is a systemic inflammatory disorder, primarily characterized by skin plaques. It is linked to co-morbidities including cardiovascular disease and metabolic syndrome. Several studies demonstrate that dietary habits can influence psoriasis development and severity. However, the effect of different dietary protein levels on psoriasis development and severity is poorly understood. In this study, we examine the influence of dietary protein on psoriasis-like skin disease in mice. Methods: We fed male C57BL/6J mice with regular, low protein and high protein chow for 4 weeks. Afterwards, we induced psoriasis-like skin disease by topical imiquimod (IMQ)-treatment on ear and back skin. The local cutaneous and systemic inflammatory response was investigated using flow cytometry analysis, histology and quantitative rt-PCR. Results: After 5 days of IMQ-treatment, both diets reduced bodyweight in mice, whereas only the high protein diet slightly aggravated IMQ-induced skin inflammation. IMQ-treatment induced infiltration of myeloid cells, neutrophils, and monocytes/macrophages into skin and spleen independently of diet. After IMQ-treatment, circulating neutrophils and reactive oxygen species were increased in mice on low and high protein diets. Conclusion: Different dietary protein levels had no striking effect on IMQ-induced psoriasis but aggravated the systemic pro-inflammatory phenotype.

## 1. Introduction

Psoriasis is one of the most common chronic inflammatory diseases of the skin, affecting 1–4% of the population worldwide [1,2]. Over a long period, psoriasis had been understood as a chronic, multifactorial, immune-mediated inflammatory skin disease hallmarked by red, scaly plaques. However, within the last decade, it has additionally been recognized as a systemic inflammatory disorder affecting the whole organism. First, psoriasis arthritis has been discovered as a well-known comorbidity in up to 40% of psoriasis patients [3], with an onset on average a decade after the cutaneous disease [4]. More recently, psoriasis has also been described as an independent cardiovascular risk factor, indicating shared inflammatory pathways and cellular mediators in skin and vessel walls [5,6].

In addition to skin, joints, and blood vessels, clinical features of metabolic syndrome, including obesity, dyslipidemia, diabetes, and nonalcoholic fatty liver disease are 40% higher in psoriasis patients compared to the general population [7,8,9,10]. Dietary and behavioral patterns may provide a simple and straightforward explanation for obesity in psoriasis patients, as the stigmatizing nature of the skin disease exerts a tremendous psychological burden on psoriasis patients [11,12]. Therefore, the well-described reciprocal link between depression and obesity [13,14] is also applicable for a significant number of psoriasis patients. However, beyond psychological explanations, different studies indicate a genetic link between psoriasis and the vicious circle of metabolic syndrome and cardiovascular disease as key driver of mortality in these patients [15]. There is evidence for single nucleotide polymorphisms [16] linking psoriasis and type 2 diabetes. Furthermore, analysis of genes associated with cardiovascular disease revealed a dysregulation of inflammatory and lipid metabolism genes in psoriasis lesions [17]. Finally, Boehncke et al. [18] developed the convincing concept of the ‘psoriatic march’: systemic inflammation in psoriasis causes insulin resistance, which in turn triggers endothelial cell dysfunction, leading to atherosclerosis and finally to cardiovascular mortality. Metabolic syndrome is not only a substantial driver of cardiovascular comorbidity in psoriasis patients. There is also evidence for a pathogenic role of obesity on the severity of the skin manifestation itself and vice versa. Different interventional studies revealed beneficial effects of weight loss on the severity of psoriasis [19,20].

Apart from weight loss, the idea of improving the course of the skin disease by dietary interventions has long been known. Already in 1913, Schamberg et al. described good results in eight patients with psoriasis on a low-protein intake [21]. They brought up the theory that retained nitrogen is used for the synthesis of epidermal protein driving hyperplastic psoriatic plaques. Therefore, protein deprivation seemed a logical step to keep the protein and therefore the nitrogen intake low and starve plaques. However, more than fifty years later, Zackheim et al. failed to reproduce any beneficial effects of a low-protein diet in a clinical study with thirteen psoriasis patients on varying levels of protein intake [22]. As they found no significant effects of different dietary protein levels on skin disease, the authors summarized that a low-protein diet is not beneficial in the management of psoriasis. To our best knowledge, this was the last interventional study focusing on different dietary protein regimes in psoriasis patients. Randomized trials or experimental data are completely missing up to now.

Today, medical organizations suggest high protein diets for increasing muscle mass and facilitating weight management [23,24]. High protein diets are highly popular and may be associated with beneficial changes in inflammatory diseases [25]. Regarding the inflammatory skin disease psoriasis, we recently provided experimental evidence that nitrogen metabolism is important for compensating water loss through inflamed skin [26]. Therefore, we now investigate the effects of different dietary protein contents in imiquimod (IMQ)-induced skin inflammation, the most popular model of psoriasis-like skin disease in mice. By repeated topical skin application of IMQ, a toll-like receptor 7/8 agonist, the animals developed erythema, scaling, keratinocyte hyper-proliferation with acanthosis and altered differentiation as well as dermal leucocyte infiltration [27]. Within several days, the mice developed an inflammatory response mimicking the hallmarks of human psoriasis.

We aimed to test the hypothesis that different dietary protein regimes can ameliorate skin and systemic inflammation in the IMQ-driven murine model of experimental psoriasis-like skin disease.

## 2. Materials and Methods

### 2.1. Mice

All mice were housed and treated in accordance with relevant laws and institutional guidelines of the Central Animal Facility of the University Medical Center Mainz, Germany. Experiments were approved by the Animal Care and Use Committee from the Land of Rhineland-Palatine, approval number G 17-1-076. Experimental procedures followed the guidelines from Directive 2010/63/EU of the European Parliament on the protection of animals used for scientific purposes.

### 2.2. Dietary Regime

Male C57BL/6J mice were obtained from Charles Rivers (Wilmington, MA, USA) at 5 weeks of age to avoid gender-related differences. The regular, high protein, and low protein diets were obtained from ssniff Spezialdiäten GmbH (Soest, Germany, Catalogue numbers V1124-300, E15202-247, E15209-347). The regular diet contained 12 kJ% fat, 27 kJ% protein, and 61 kJ% carbohydrates (nitrogen-free-extracts 51.2%). The high protein diet contained 20 kJ% fat, 49 kJ% protein, and 31 kJ% carbohydrates (nitrogen-free extracts 29.5%). The low protein diet contained 19 kJ% fat, 9 kJ% protein, and 72 kJ% carbohydrates (nitrogen-free-extracts 69.7%). Composition of all diets is displayed in Figure 1A. The mice received a high protein, low protein or regular diet ad libitum for 4 weeks starting at the age of 5 weeks. We monitored bodyweight and food consumption twice a week. For calculating caloric uptake, the consumed food per mouse in gram was multiplied by the calories per gram of the respective type of food.

### 2.3. Imiquimod-Induced Psoriasis-Like Skin Disease

Psoriasis-like skin disease was induced with imiquimod (IMQ) treatment of ear and back skin. 9-week-old mice were shaved and depilated (Balea, dm, Karlsruhe, Germany) on their backs 24 h prior to treatment. Mice received treatment with Aldara cream (Meda AB, Solna, Sweden) containing 5% of IMQ or sham cream (Vaseline-Lanette-Crème. Pharmacy of the University Medical Center Mainz, Mainz, Germany) on their back skin (50 mg) and ears (5 mg each) once daily for five consecutive days. After five days of either IMQ or sham treatment mice were deeply anesthetized with isoflurane and sacrificed by total blood collection by cardiac puncture. 

### 2.4. Psoriasis Area and Severity Scoring (PASI)

During treatment, the severity of IMQ-induced psoriasis was scored daily using a modified PASI score consisting of the parameters scaling, erythema and ear thickness. By using a caliper (µm) the thickness of ears was measured and erythema and scaling were scored (score range = 0–4). The modified cumulative PASI is calculated as scaling score + erythema score + ear thickness change (%) [28,29].

### 2.5. Flow Cytometry

For cell preparation, spleens were isolated, mechanically pressed through a nylon cell strainer (40 µm), and flushed with PBS/FCS 2% (*v*/*v*). Red blood cells were lysed by adding 1× ammonium/chloride/potassium (ACK) buffer (1.5 M NH_4_Cl, 100 mM KHCO_3_, 10 mM EDTA-2Na) for 3 min at room temperature. Ear skins were disrupted mechanically, digested with 0.15 mg/mL Liberase and 0.12 mg/mL DNase I (Roche, Basel, Switzerland) for 60 min at 37 °C. Afterwards, the cells were filtered through a 70 µm cell strainer to obtain a single-cell solution. Spleen and ear skin cells were centrifuged (300× *g*, 6 min, 4 °C) and cells were counted (Spark^®^ Multimode Microplate Reader; Tecan, Maennedorf, Switzerland).

To block unspecific binding sites, cells were incubated with Fc-block (Bio X Cell, Lebanon, NH, USA). The following monoclonal antibodies were used for cell surface staining: CD45.2 (clone: 104, BV506, Biolegend, San Diego, CA, USA or APC-Cy7, BD Bioscience, Franklin Lakes, NJ, USA), CD11b (clone: M1/70, PE-Cy7, eBioscience, San Diego, CA, USA), Ly6G (clone: 1A8, SB600 or PE, BD Bioscience, Franklin Lakes, NJ, USA), Ly6C (clone: AL-21, V450, eBioscience, San Diego, CA, USA), F4/80 (clone: BM8, APC, eBioscience, San Diego, CA, USA). CD45R/B220 (clone: RA3-6B2, PerCP eFluor 710, BD Biosciences, Franklin Lakes, NJ, USA).

Samples were acquired using FACSCanto™II (BD, Franklin Lakes, NJ, USA) or Attune™ NxT (Thermo Fisher, Waltham, MA, USA) Flow Cytometer and analyzed using FlowJo Software (BD, Franklin Lakes, NJ, USA).

### 2.6. Blood Count

To quantify peripheral blood counts, we used the VetScan HM5 (Abaxis, Union City, CA, USA) a fully automated veterinary hematology analyzer reporting a complete blood count (CBC) using whole blood.

### 2.7. ROS/RNS Measurement

Reactive oxygen and nitrogen species (ROS/RNS) were analyzed in whole blood. In order to stimulate the formation of an oxidative burst in leukocytes, the blood was incubated for 20 min using phorbol 12,13-dibutyrate (PDBu). The oxidative burst was measured by 8-amino-5-chloro-7-phenylpyridol (3,4-d)pyridazine-1,4-(2H,3H) dione sodium salt (L-012)-enhanced chemiluminescence as described before [30,31].

### 2.8. Quantitative Real Time PCR

For RNA isolation, liver and skin tissue was pulverized or homogenized using the TissueLyser II (Quiagen, Hilden, Germany) and suspended in Git buffer (4 M guanidinium-isothiocyanate, 25 mM Na-citrat, 0.5% N-lauroylsarcosine, 7.2% β-mercaptoethanol). RNA was extracted based on the phenol-chloroform extraction protocol [32]. RNA concentration was measured with a NanoDrop™ spectrophotometer (Thermo Fisher, Waltham, MA, USA) or Tecan’s Spark^®^ NanoQuant Plate™ (Tecan Spark, Tecan Inc., Maennedorf, Switzerland). 0.5 µL of total RNA was used for a TaqMan^®^ Gene Expression Assay (Applied Biosystems™, Waltham, MA, USA) as described in the manufacturer’s protocol by using the following primers: *IL-6* (Mm00446190_m1), *TNFα* (Mm00443260_m1), *CXCL2* (Mm00441242_m1) and *TATA-box* (Mm00446973_m1).

### 2.9. Histology

For hematoxylin and eosin staining, tissues were dissected and fixed in 4% paraformaldehyde, paraffin-embedded, cut, and stained with hematoxylin and eosin according to standard protocols. To determine epidermal thickness, 10 consecutive images per animal were taken using an Olympus IX73 microscope with an Olympus SC30 camera (Olympus, Tokyo, Japan).

### 2.10. Statistics

Data are displayed as mean ± standard error of the mean (SEM). Statistical calculations were performed with GraphPad Prism software (version 9; GraphPad Software Inc., San Diego, CA, USA). Data were analyzed for normal distribution (Kolmogorow–Smirnow test). When normal distribution was given, we applied the one-way or 2-way ANOVA test with Tukey’s post-hoc test. If no normal distribution was given, a Kruskal–Wallis test with Dunn’s multiple comparison or comparison of selected columns was used as appropriate and indicated in the figure legends. *p*-values of < 0.05 were considered significant and marked by asterisks (* *p* < 0.05; ** *p* < 0.01; *** *p* < 0.001).

## 3. Results

### 3.1. Four Weeks of Protein Diet Altered Neither Calorie Intake nor Bodyweight in C57BL/6J Mice

Before inducing psoriasis-like skin disease, mice were fed with regular, low, or high protein chow for four consecutive weeks starting at the age of 6 weeks. The regular diet consisted of 27 kJ% protein, compared to the low protein diet which only contained 9 kJ% protein but 7 kJ% more fat as well as 11 kJ% more carbohydrates (Figure 1A) resulting in an increased energy content of 16.2 MJ/kg compared to regular diet containing 14 MJ/kg. Compared to the low protein diet, the high protein diet contained an excessive amount of 49 kJ% protein resulting in a more than 40 kJ% lower concentration of carbohydrates but a comparable amount of fat to achieve an energy content of 15.8 MJ/kg.

To exclude differences in food intake, we measured and calculated food intake per mouse for all dietary regimes every three to four days. Starting from the beginning, food intake was significantly lower in both groups fed with altered protein content (Figure 1B). Given the higher energy content of the low and high protein diet, the calculated calorie intake was comparable without significant differences between the three groups during the four weeks of feeding (Figure 1C). As expected, all mice showed a steady increase in bodyweight over time. After 4 weeks, we did not find any differences in bodyweight gain between the experimental groups (Figure 1D).

### 3.2. High Protein Intake Lowers Bodyweight and Affects Skin Inflammation in IMQ-Induced Psoriasis-Like Skin Disease without Altering Cutaneous Infiltration of Myeloid Cells

After four weeks of feeding, we initiated the skin treatment with cream containing 5% IMQ or sham cream (graphic presentation of the experimental setting in Figure 2A).

Daily recordings of the bodyweight showed that sham treatment did not affect bodyweight during the entire 5-day course, whereas all IMQ-treated mice significantly lost bodyweight after the second day of treatment (Figure 2B). During the first 4 days, there were no differences regarding bodyweight between the IMQ-treated animals on regular, low or high protein diets. Notably, on the fifth (and last) day of IMQ-treatment, IMQ-treated animals on a high protein diet had a significantly lower bodyweight compared to IMQ-treated animals on a regular diet whereas the animals receiving regular or low protein diets did not show any differences in bodyweight (Figure 2B).

As described before, IMQ-treatment induced erythema, scaling and skin swelling. Different dietary protein levels had no significant effects on this finding (Figure 2C and Supplemental Appendix A). The combined cumulative PASI (consisting of skin thickening, scaling, and erythema) also increased during the treatment period, without differences between the three dietary groups. Again, on the final day of IMQ-treatment, mice on a high protein diet seemed to exhibit a stronger skin phenotype than the IMQ-treated animals on regular protein chow (Figure 2C). In H&E-stained skin sections, we found a highly thickened epidermal layer in all dietary groups compared to sham-treated animals on a regular diet. Alterations in dietary protein levels did not alter this effect of IMQ on the epidermal diameter (Figure 2D,E). Furthermore, we studied the mRNA content of *tumor necrosis factor alpha* (*TNF-α*), *Interleukin-6* (*IL-6*) and *C-X-C motif chemokine ligand 2* (*CXCL2*) in the skin tissue of sham or IMQ-treated skin from mice on all three different diets. We did not find changes in *IL-6* and *CXCL2* expression but found a significant induction of *TNF-α* in IMQ-treated mice a on low protein diet compared to untreated mice whereas IMQ-treatment in animals on other diets failed to increase *TNF-α* (Figure 2F).

As a consequence of IMQ-treatment, we found an infiltration of myeloid cells into the skin as previously demonstrated [28]. There was no difference in the infiltration of CD11b^+^ myeloid cells in the skin in sham treated animals of all dietary groups (Figure 3A,B). IMQ-treatment resulted in elevated levels of dermal CD11b^+^ cells (Figure 3A,B) and neutrophils (CD11b^+^Ly6C^+^Ly6G^+^) in all groups (Figure 3C,D), whereas the percentage of skin monocyte/macrophage (CD11b^+^Ly6G^−^Ly6C^low and high^) counts did not significantly increase under IMQ-treatment (Figure 3E).

### 3.3. High Protein Diet Increases Circulating Neutrophils and Reactive Oxygen Species in Imiquimod-Induced Skin Inflammation

Besides local skin inflammation, cutaneous IMQ-treatment has been shown to exert systemic pro-inflammatory capacities [33]. We therefore analyzed myeloid cells in the spleen and blood. Additionally, we further identified Ly6C^high^ monocytes/macrophages, which are described to exert a pro-inflammatory phenotype (Appendix A for representative FACS plots of the gating strategy). Comparable to the skin, there was an increase of CD11b^+^ cells after IMQ-treatment in the spleen. We did not find any differences between regular, low and high protein dietary regimes (Figure 4A,B). There was an increase in splenic neutrophils (CD11b^+^Ly6C^+^Ly6G^+^) in all IMQ-treated mice—which was significant in IMQ-treated animals on a high protein diet (Figure 4C,D) only. Increase of splenic monocytes/macrophages (CD11b^+^Ly6G^−^Ly6C^low and high^) under IMQ was not influenced by our protein diets (Figure 4E). We found the same trend for the splenic Ly6C^high^ pro-inflammatory monocytes/macrophages (Figure 4F).

IMQ-treatment did lower total leucocyte blood counts by trend without reaching significance (Figure 5A). This finding was driven by a significant reduction in circulating lymphocytes (Figure 5B) in IMQ-treated animals independently of protein intake. We detected higher levels of circulating neutrophils in IMQ-treated compared to sham-treated animals of all diets. IMQ-treated animals on a regular diet had significantly lower neutrophil counts than IMQ-treated animals on a low or high protein diet (Figure 5C). Finally, we assessed the production of reactive oxygen species in whole blood by L-012 enhanced chemiluminescence (ECL) stimulated with PDBu. Consistent with the detected neutrophil counts, IMQ-treated mice on high and low protein diets had significantly higher peripheral ROS/RNS levels than IMQ-treated animals on a regular diet (Figure 5D).

## 4. Discussion

In this study, we demonstrate that dietary modulation of protein intake has little impact on IMQ-induced psoriasis-like skin disease but nevertheless impacts systemic inflammation. The association between diet and the risk of developing inflammatory autoimmune diseases such as rheumatoid arthritis [34,35], multiple sclerosis [36], systemic lupus erythematosus [37], or inflammatory bowel disease [38] was proposed more than 50 years ago. The widespread “Western diet” (high-fat and cholesterol, high-sugar, and excess salt intake) not only drives obesity, metabolic syndrome and cardiovascular disease, but has also been suspected as a crucial driver for almost all autoimmune diseases [39]. De Rosa et al. speculated that increased caloric intake leads to an altered adipose tissue homeostasis with increased synthesis of adipokines [40]. Along their theory, this ‘metabolic pressure’ of over-nutrition results in low-grade inflammation and activation of pro-inflammatory immune-cells by over-activation of nutrient-sensing mechanisms and ultimately results in the loss of immunotolerance.

Different studies have revealed that dietary interventions relying on high fat intake aggravate skin inflammation in experimental psoriasis. Zhang et al. demonstrated that mice fed a high fat diet develop dermatitis without further additional stimulus. Moreover, animals on high fat suffered from more severe IMQ-driven skin inflammation with increased numbers of CD11c^+^ macrophages in the skin [41]. In addition to macrophages, a high fat diet also exacerbated the number of IL-17-producing γδ-T cells [42] and neutrophils [43] in the skin after IMQ-treatment. We did not find major changes in the IMQ-induced skin alterations between the dietary groups. Although the cumulative PASI score was slightly increased in mice on a high protein diet on the last day of IMQ-treatment, we detected differences neither in skin thickness, nor in myeloid cell infiltration in the skin. Our data strongly indicates that altering protein intake over four weeks does not exert comparable detrimental effects to increasing dietary fat on skin infiltration of pro-inflammatory cells or psoriasis severity. At the end of our pre-treatment feeding period of four weeks, mice successfully adapted to the changes in diet. Despite a significantly lower amount of ingested food, the animals on high or low protein diets had the same bodyweight after four weeks of feeding as the control group due to the higher calorie-content of the experimental diets. This is in line with previous publications focusing on low or high protein diets in different experimental settings [44,45,46] and reveals another big difference to the studies based on high fat diets, as animals fed high fat before IMQ-treatment almost doubled their bodyweight [42]. In our study, IMQ-treatment resulted in a significantly lower bodyweight in the high protein group compared to animals on a regular diet. This finding might suggest a negative impact of the diet on the general health of the mice.

Nevertheless, under low and high protein diets, we found increased neutrophils in the blood where reactive oxygen species were also elevated. Thus, both diets in fact pushed the systemic immune situation towards a more reactive phenotype—perhaps as the organism struggles with a new situation of nutritional supply. In contrast to our findings, it was shown that overall inflammation and oxidative stress generally increased less in participants of the Framingham Heart Study Offspring with the highest protein intake, hinting to possible support for anti-inflammatory processes by a high protein diet [25]. The question of whether long-term rearrangement of protein intake might have a beneficial effect in human psoriasis needs to be analyzed in further studies, especially in combination with existing therapeutic options.

There are limitations regarding our study that require consideration: Studies using animal models of psoriasis have consistently increased over the past four decades, but all available experimental models only imperfectly mimic the complexity of the pathogenesis seen in human patients [47,48,49]. Since 2009, IMQ-induced psoriasis-like skin disease represents the most popular model to study psoriasis-like inflammation in mice. Almost 50% of all publications using animal models of psoriasis were based on this model [50], even if the immunopathology differs from human psoriasis. The legitimate question of whether effects or the absence of effects in this model can be translated to human patients is a strong limitation of all studies based on experimental data in this model—including our present study. Furthermore, the applied diets not only differed in protein but also in carbohydrates and fat to allow similar energy contents, so the observed changes cannot be attributed to protein alone.

## 5. Conclusions

Nutrition is one essential factor in the management of various inflammatory diseases. However, the complexity of the interaction between nutrition and immunology is immense [51]. Psoriasis, which is one of the most common inflammatory skin diseases worldwide, does not represent an exception for this complex field [52]. In conclusion, we show that neither low nor high protein diets had striking effects on skin inflammation in IMQ-induced psoriasis but both nutrition regimens increased oxidative stress levels and circulating neutrophils in blood.

## Figures and Tables

**Figure 1 nutrients-13-01897-f001:**
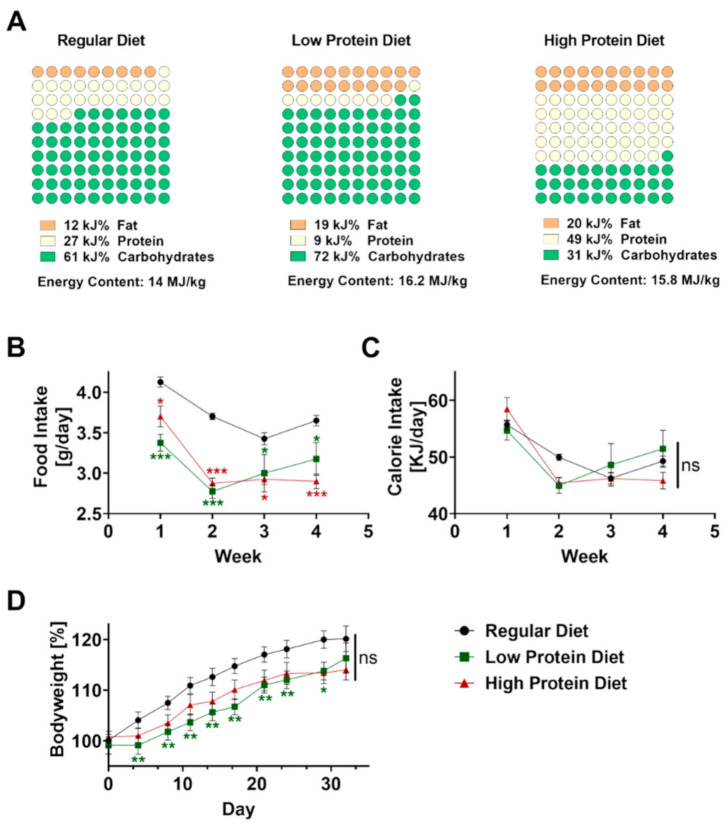
Effects of feeding a regular, low or high protein diet on food intake and bodyweight of C57BL/6J mice. (**A**) Composition of the dietary regimes. Percentage of fat, protein and carbohydrates as well as energy content. (**B**) Food intake of mice on a regular, low or high protein diet during the period of four weeks (*n* = 4; 2-way ANOVA). (**C**) Calculated calorie intake during the feeding period of the same animals (*n* = 4, 2-way ANOVA). (**D**) Change in bodyweight of the same animals during the period of four weeks on regular, low or high protein diet (*n* = 30–32; 2-way ANOVA). Data are shown as mean ± SEM. Green asterisks show significant differences between control and low protein diet fed mice, red asterisks between control and high protein diet. *** *p* < 0.001; ** *p* < 0.01; * *p* < 0.05.

**Figure 2 nutrients-13-01897-f002:**
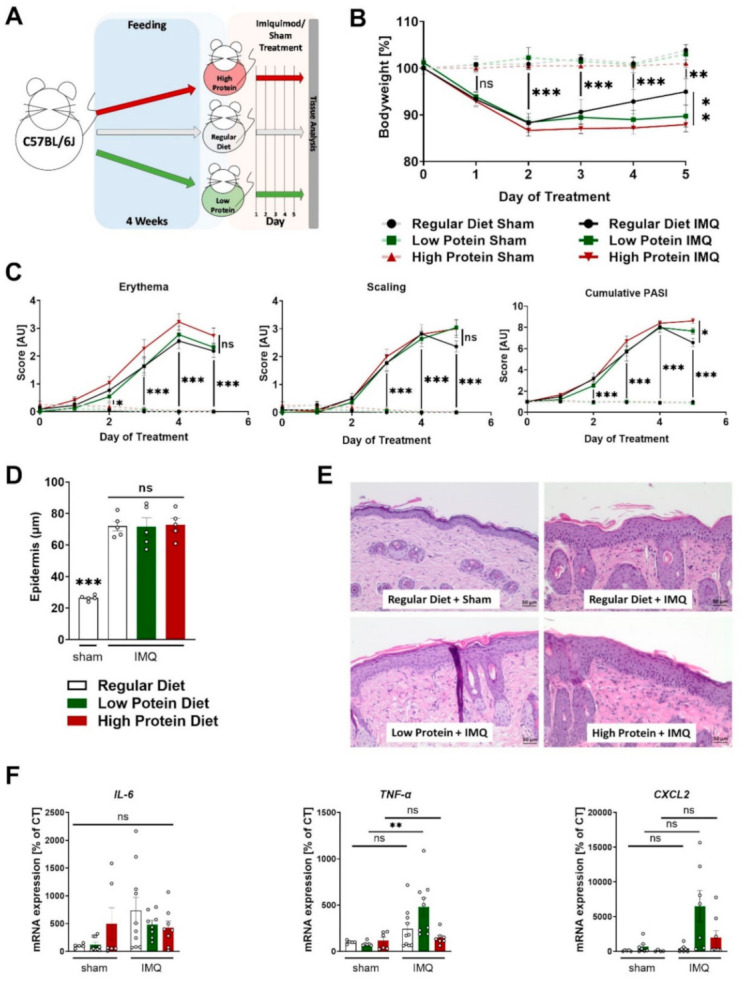
Modulating dietary protein only slightly affects skin inflammation in IMQ-induced psoriasis-like skin disease. (**A**) Graphic presentation of the experimental setting. (**B**) Change in bodyweight during treatment in sham or IMQ-treated animals on a regular, low, or high protein diet (*n* = 4–13; 2-way ANOVA. (**C**) Scoring of the severity of skin disease during sham or IMQ-treatment in the different dietary groups. Erythema and scaling were graded, and cumulative PASI-score calculated (*n* = 8–13; 2-way ANOVA). (**D**) Quantification of the thickness of the epidermal layer using H&E-stained skin sections from sham treated animals on a regular diet or IMQ-treated animals on a regular, low, or high protein diet (*n* = 5; 1-way ANOVA). (**E**) Representative H&E-stained skin sections from a sham treated mouse and lesional skin of IMQ-treated animals on different diets. Scale bars equal 50 µm. (**F**) Expression of *IL-6*, *TNF-α*, and *CXCL2* mRNA in skin of sham or IMQ-treated animals on regular, low, or high protein diet (*n* = 5–10; 1-way ANOVA). Data are shown as mean ± SEM. Black asterisks show significant differences between sham and IMQ-treated mice. *** *p* < 0.001; ** *p* < 0.01; * *p* < 0.05.

**Figure 3 nutrients-13-01897-f003:**
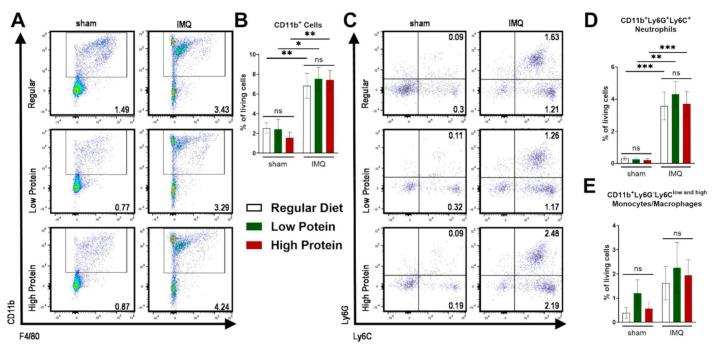
IMQ-treatment drives skin infiltration of myeloid cells independently of the dietary protein regime. Flow cytometric analysis of ear skin after sham or IMQ-treatment in mice on a regular, low, or high protein diet. (**A**) Representative flow cytometric plots gated on CD11b and F4/80 are shown for each group. (**B**) Quantification of CD11b^+^ myeloid cells in the skin. To determine CD11b^+^ myeloid cells, cells were pre-gated on living CD45.2^+^ cells (*n* = 4–13; Kruskal–Wallis test). (**C**) Representative flow cytometric plots gated on Ly6G and Ly6C are shown for each group. Quantification of skin (**D**) neutrophils (CD11b^+^Ly6G^+^Ly6C^+^) and (**E**) monocytes/macrophages (CD11b^+^Ly6G^−^Ly6C^low and high^) (*n* = 4–13; Kruskal–Wallis test). Data are shown as mean ± SEM. *** *p* < 0.001; ** *p* < 0.01; * *p* < 0.05.

**Figure 4 nutrients-13-01897-f004:**
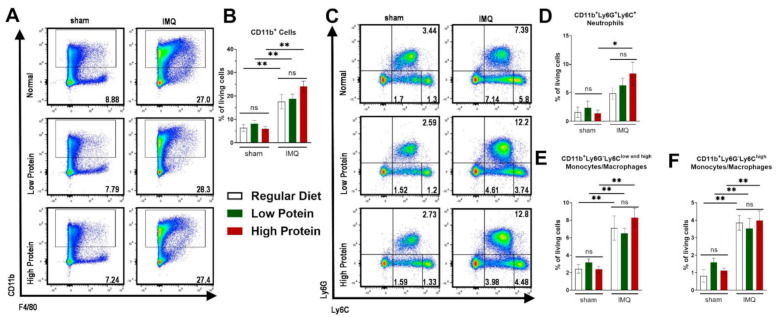
Modulation of dietary protein has no effects on IMQ-driven expansion of myeloid cells in the spleen. Flow cytometric analysis of spleen cells after sham or IMQ-treatment in mice on a regular, low, or high protein diet. (**A**) Representative flow cytometric plots of spleen cells from sham or IMQ-treated-mice on a regular, low, or high protein diet. (**B**) Quantification of CD11b^+^ myeloid cells. To determine CD11b^+^ myeloid cells, cells were pre-gated on living CD45.2^+^ cells (*n* = 4–13; Kruskal–Wallis test). (**C**) Representative flow cytometric plots gated on Ly6G and Ly6C are shown for each group. (**D**) Quantification of CD11b^+^Ly6G^+^Ly6C^+^ neutrophils (*n* = 4–13; Kruskal–Wallis test). (**E**) Quantification of CD11b^+^Ly6G^−^Ly6C^low and high^ monocytes/macrophages and (**F**) the subset of CD11b^+^Ly6G^−^Ly6C^high^ monocytes/macrophages in the spleen after sham or IMQ-treatment in mice on regular, low, or high protein diet (*n* = 4–13; Kruskal–Wallis test). Data are shown as mean ± SEM. ** *p* < 0.01; * *p* < 0.05.

**Figure 5 nutrients-13-01897-f005:**
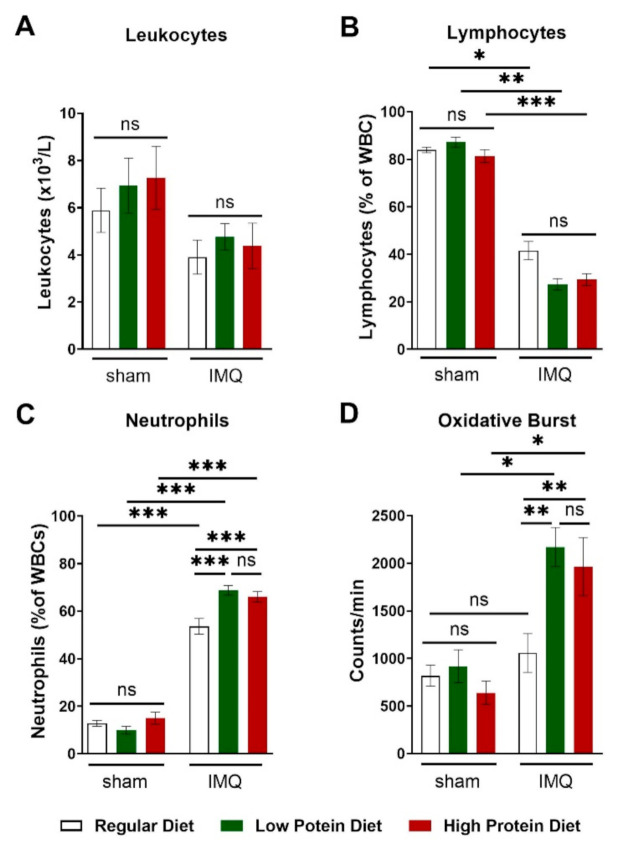
High and low protein diets increased numbers of circulating neutrophils and boosted the formation of reactive oxygen species in IMQ-induced psoriasis-like skin disease. (**A**) Absolute number of blood leucocyte counts in sham or IMQ-treated mice on a regular, low, or high protein diet (*n* = 8–13; 2-way ANOVA). (**B**) Relative amount of blood lymphocytes and (**C**) neutrophils presented as % of white blood cells in sham or IMQ-treated mice on a regular, low, or high protein diet (*n* = 8–13; 2-way ANOVA). (**D**) ROS/RNS measurement in whole blood (20 min of PDBu stimulation), repeated measurements of pooled samples; (*n* = 5–12; 2-way ANOVA). Data are shown as mean ± SEM. *** *p* < 0.001; ** *p* < 0.01; * *p* < 0.05.

## Data Availability

All data generated or analyzed during this study are included in the manuscript (and its supplementary information files).

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
