# Peer review of "Effects of Dietary Protein Intake on Cutaneous and Systemic Inflammation in Mice with Acute Experimental Psoriasis"

_nutrients, 2021, doi:10.3390/nu13061897_

Round 1

Reviewer 1 Report

The paper attempts to address the gap of whether dietary protein is of relevance when it comes to inflammatory conditions such as psoriasis. The murine model used is well characterised for the human condition.

I find it hard to correlate the chnages observed in the model solely to the 'protein' content of the diet. The mouse diets used were limited in the sense that the amounts of carbohydrate and fat was not controlled, so in essence a "high protein diet" was also a 'high fat and low carbohydrate' diet and a "low protein diet" was also a 'high carbohydrate and high fat diet'. Despite the energy content conversion to standardise the diet, the main assumption of the study stated that it is the nitrogen content derived from protein diet (and not the energy content) that leads to exacerbation of inflammatory disease. Hence, the overall assumption of the study is not represented by the methodology used. The study could measure the nitrogen levels of the diet to justify the definition of high/low/normal protein. At present, the justification of all inflammatory markers and changes to the mouse cells can be equally attributed to carbohydrates and fat. Despite this obvious conundrum, the study is sound and the data is worth publishing - however, the results cannot be attributed to protein alone. I urge the authors to perhaps alter the title and remove the overemphasis on obesity (fat diet) to psoriasis if they will not address the differences in the fat content of the diets. The authors could also note that reduction on body weight in mice is a sign of ill-health. 

The supplementary material was not available for review.

Minor corrections:

  1. page 1, line 34. remove 'today' to make sentence read better.
  2. page 2, line 50. Analysis not analyses as reference is to a single study/paper.
  3. page 2, line 57. add "and" vice versa. 
  4. page 2, line 60. "has long been known" instead of 'has a long tradition'.
  5. page 2, line 72. Clarify and state which other organisations promote high protein diets.
  6. page 2, line 75/76. "we recently provided experimental evidence that nitrogen metabolism is important for compensating water loss...".
  7. page 2, line 77. "we now investigate the effects of..."
  8. page 2, line 78, remove 'murine' - repeated with mouse in next line.
  9. page 2, line 83. "mimicking the hallmarks of..."
  10. page 3, line 99-101. ensure explanation offered for difference in fat% between diets - thes have chnaged significantly. Therefore the altered diets vary in both protein and fat. Explained in Figure 1 A - thank you for the schematic; very easy to visualise proportions/percentage protein and fat.
  11. page 3, line 125. skins (plural)
  12. page 3, line 128. spleen and ear skin "cells" were centrifuged...
  13. page 4, line 145. measured typo.
  14. page 4, line 150. Git buffer composition?
  15. page 4, line 173/174. four weeks of protein diet altered neither...
  16. page 7. Figure 3A. can authors state that F4/80 marker can be used to distinguish between CD11b monocytes versus CD11b macrophages?
  17. page 8. Figure 4A. In the facs plots, along the F4/80 axis, there is a population of cells at 45 degree angle that is higher in normal diet mice, and still present in high protein diet, but almost disappeared in low protein diet. These F4/80 high cells - have the authors measured this separately to link differences to the pro-inflammatroy macrophage phenotype? 
  18. page 8, line 275. sentence unclear. Clarify whether normal diet netrophils were same as or higher than other diets. 
  19. page 9, Figure 5. typo. increase = increased.
  20. page 10, line 312. typo. indicate = indicates.
  21. page 10, line 315. remove "of" to read "despite a significantly..."
  22. page 10, line 323. "the blood where reactive oxygen species were also elevated"

Author Response

We would like to thank the reviewer for the helpful comments. Please find our point-by-point response to the reviewers' comments in the attached word file.

Reviewer 2 Report

In this study, the authors challenge the idea that a low-protein diet can improve psoriasis by decreasing inflammation. They fed mice with 3 different regimens; normal, low-, and high-protein diets; then induced psoriasiform skin lesions by applying imiquimod. They found no differences in the development of skin psoriasis between the 3 diets yet they observed an increased circulation of neutrophils and ROS production that may promote systemic inflammation.

The study is conducted correctly and well described however some conclusions seem incorrect for the following reasons:

1/ In Fig.2F, it is hard to believe there is no increased expression of CXCL2 but only TNF in the skin of low-protein fed mice. How many mice are used here? the legend says n=5-10, could the authors plot the data points of individual mice to better see the distribution? If not statistically significant, the authors could measure the expression of CXCL1 instead.

2/ In all figures showing dot plots, the authors should add the percentages of their gates.

3/ In Fig.3A, the CD11b+ F4/80+ population seems more important in the sham-treated mice with regular diet than with the other diets. Moreover, in the IMQ-treated mice, this population that is present in the regular diet disappears in the other diets. The authors should quantify this F4/80+ macrophage population.

4/ Still in Fig.3A, a CD11b*high F4/80- population appears in the skin of IMQ-treated mice fed with low- and high-protein diets that is not present in the regular diet. The authors should quantify this population instead of quantifying "total" CD11b+ cells.

5/ In Fig.4A, same as in Fig.3A, the quantification of CD11b+F4/80- versus CD11b+F4/80+ cells should be performed instead of total CD11b+ cells.

6/ Why the authors choose to perform experiments only with male mice?

Author Response

(The authors gave the same response as above.)

Round 2

Reviewer 2 Report

The authors performed the analyses I requested, which did not change the conclusions.

Although I also asked for depicting each data point in the bar graphs, this was not done by the authors.